# Combining mTOR Inhibitors and T Cell-Based Immunotherapies in Cancer Treatment

**DOI:** 10.3390/cancers13061359

**Published:** 2021-03-17

**Authors:** Alexandre el Hage, Olivier Dormond

**Affiliations:** Department of Visceral Surgery, Lausanne University Hospital, University of Lausanne, 1011 Lausanne, Switzerland; alexandre.elhage@unil.ch

**Keywords:** cancer, mTOR, rapalogs, immunotherapy

## Abstract

**Simple Summary:**

Several clinical protocols are exploring the anticancer effect of immunotherapy combined to targeted therapy. Indeed, emerging evidence demonstrates that small-molecule targeted inhibitors promote the antitumor immune response. In the case of mechanistic target of rapamycin (mTOR) inhibitors, such approach appears initially counterintuitive due to their immunosuppressive properties. Nevertheless, recent studies have highlighted the complex role played by mTOR in immune cell biology and have demonstrated that mTOR inhibitors can promote or repress the immune response in a context-dependent manner. Accordingly, pre-clinical studies have shown that mTOR inhibitors have the potential to increase the antitumor efficacy of immunotherapies. Here, we therefore review the different inhibitors of mTOR and their effects on adaptive immunity related to current immunotherapies. We further discuss the therapeutic opportunities of combining mTOR inhibitors with immunotherapies in cancer treatment.

**Abstract:**

mTOR regulates several processes that control tumor development, including cancer cell growth, angiogenesis and the immune response to tumor. Accordingly, mTOR inhibitors have been thoroughly explored in cancer therapy but have failed to provide long-lasting anticancer benefits. Several resistance mechanisms that counteract the antitumor effect of mTOR inhibitors have been identified and have highlighted the need to use mTOR inhibitors in combination therapies. In this context, emerging evidence has demonstrated that mTOR inhibitors, despite their immunosuppressive properties, provide anticancer benefits to immunotherapies. In fact, mTOR inhibitors also display immunostimulatory effects, in particular by promoting memory CD8^+^ T cell generation. Hence, mTOR inhibitors represent a therapeutic opportunity to promote antitumor CD8 responses and to boost the efficacy of different modalities of cancer immunotherapy. In this context, strategies to reduce the immunosuppressive activity of mTOR inhibitors and therefore to shift the immune response toward antitumor immunity will be useful. In this review, we present the different classes of mTOR inhibitors and discuss their effect on immune cells by focusing mainly on CD8^+^ T cells. We further provide an overview of the different preclinical studies that investigated the anticancer effects of mTOR inhibitors combined to immunotherapies.

## 1. Introduction

The mechanistic target of rapamycin (mTOR) regulates several biological mechanisms implicated in tumor progression, including cancer cell proliferation and tumor angiogenesis [1,2]. Accordingly, mTOR inhibitors have been intensively studied as potential anti-cancer agents. However, despite promising pre-clinical studies, they have failed to provide long-lasting benefits in cancer patients. Several resistance mechanisms developed by tumors to overcome mTOR inhibition have been identified and have helped explain the modest anticancer efficacy of mTOR inhibitors [3]. Besides resistance mechanisms, one other major concern regarding the efficacy of mTOR inhibitors relates to their immunosuppressive properties. Indeed, rapamycin, a first generation inhibitor of mTOR, is used to prevent organ rejection in transplanted patients [4]. Consequently, the immunosuppressive effect of mTOR inhibitors could counteract their anticancer efficacy. Interestingly, studies from the last decade have demonstrated that the role of mTOR in adaptive immunity is complex and can in fact either promote or dampen the immune response [5]. In this context, emerging evidence has shown that the antitumor effects of mTOR inhibitors are partly mediated by the adaptive immune response. For instance, decreased tumor growth induced by rapamycin is abrogated by CD8^+^ T cell depletion in mice bearing MOC1 tumors [6]. Similarly, growth reduction of invasive lobular carcinoma of the breast by mTOR inhibition is significantly decreased in Rag1^−/−^ mice lacking mature T and B lymphocytes compared to immunocompetent mice [7]. Importantly, the paradoxical effect of mTOR inhibitors on immunity has also been observed in cancer patients where the rapamycin analog everolimus simultaneously promoted high expansion of regulatory CD4^+^ T cells and activated tumor-specific Th1 immunity [8]. Hence, whereas mTOR inhibitors were classically considered as immunosuppressive agents, recent observations clearly suggest that they act rather as immunomodulators. Therefore, identifying conditions that promote the immunostimulatory effects of mTOR inhibitors over the immunosuppressive ones has the potential to provide therapeutic benefits to immunotherapies. This review aims to present mTOR inhibitors and their effect on adaptive immunity and to discuss the anticancer potential of combining mTOR inhibitors with different modalities of cancer immunotherapy. A particular emphasis is given to the experimental protocols since the design of appropriate approaches regarding drug regimen is critical. In particular, dose and administration schedule will be important as degree and timing of mTOR inhibition positively or negatively impact the anti-tumor immune response.

## 2. mTOR Inhibitors

mTOR is a serine/threonine kinase ubiquitously expressed and present in three distinct protein complexes, named mTORC1, mTORC2 and the recently described mTORC3 (Figure 1) [9,10]. mTOR contains huntingtin, elongation factor 3, protein phosphatase 2A and TOR1 (HEAT) repeats at its N-terminal followed by a FRAP, ATM, TRRAP (FAT) domain, a FKBP12-rapamycin binding (FRB) domain, a kinase domain and a domain at the C-terminus (FATC). mTOR binds via these different domains to core components of the complexes such as DEP-domain-containing mTOR-interacting protein (DEPTOR) and mammalian lethal with SEC13 protein 8 (mLST8) that interact with the FAT and kinase domains respectively, and that are present in both complexes [11,12]. In addition, mTOR binds regulatory-associated protein of mTOR (RAPTOR) or rapamycin-insensitive companion of mTOR (RICTOR) at its HEAT repeats, the defining subunits of mTORC1 and mTORC2 [13,14,15,16]. In turn, RAPTOR recruits the mTORC1 inhibitory protein proline-rich AKT substrate 40kDa (PRAS40) and RICTOR recruits protein associated with RICTOR 1 or 2 (PROTOR1/2) and MAPK interacting protein 1 (mSIN1) to mTORC2 [17,18,19,20,21]. In contrast to mTORC1 and mTORC2, little is known regarding the composition of mTORC3. In this complex, mTOR interacts with ETS variant transcription factor 7 (ETV7) and stimulates cell proliferation [10]. Finally, the description of an mTOR/G-protein-coupled receptor kinase-interacting protein 1 (GIT1)/PAK-interacting exchange factor (β-PIX) complex that regulates astrocytes’ survival suggests that other mTOR-containing complexes exist [22].

Besides their different protein composition, mTORC1 and mTORC2 have distinct cellular functions. mTORC1 senses the environmental milieu, and in the presence of favorable conditions, stimulates cellular processes that lead to biomass accumulation and cell growth [9]. Thus, whereas oxygen, energy, nutrients and growth factors activate mTORC1, stress such as acidity represses it. Once activated, mTORC1 upregulates protein, lipid and nucleotide synthesis and inhibits autophagy. mTORC2 is mainly activated by growth factors and promotes proliferative and pro-survival pathways and participates in cytoskeletal rearrangements [9].

Several chemical inhibitors exist to target mTOR complexes. In the 1970s, the macrocyclic lactone rapamycin was isolated from the soil bacterium *Streptomyces hygroscopicus* [23]. Nearly 20 years later, its mechanism of action was elucidated and mTOR was identified as a kinase inhibited by rapamycin [24,25,26]. At the molecular level, rapamycin, associated to the prolyl-isomerase FKBP12, binds the FRB domain in mTOR and inhibits mTORC1 allosterically [27,28]. Since the FRB domain is unique to mTOR, rapamycin is selective for mTOR (Table 1). This inhibition is only partial as several protein residues phosphorylated by mTORC1 are resistant to rapamycin [29,30]. Nevertheless, high doses of rapamycin, micromolar instead of nanomolar concentrations, provide a more profound inhibition of mTORC1 activity in a FKBP12-independent but FRB domain-dependent mechanism [31]. mTORC2 is also sensitive to high doses of rapamycin [31,32]. Analysis of the cryo-electron microscopy reconstruction of mTORC1 further demonstrated that the FKBP-12-rapamycin complex partially obstructs the access of substrate to the kinase active site of mTOR [33]. Similar analysis for mTORC2 revealed that RICTOR blocks the FKBP-12-rapamycin binding site on mTORC2, thus explaining the lack of direct inhibition of mTORC2 by rapamycin [34]. However, in certain cell types, chronic exposure to rapamycin still inhibits mTORC2, presumably by blocking de novo formation of mTORC2 as FKBP-12-rapamycin complex sequesters a pool of mTOR necessary for mTORC2 assembly [35]. Whether prolonged rapamycin treatment similarly blocks mTORC1 de novo formation and therefore provides full mTORC1 inhibition has not been reported. Finally, rapamycin does not inhibit mTORC3 [10].

The immunosuppressive and anticancer properties of rapamycin were reported soon after its isolation, highlighting a substantial clinical potential [36,37]. Accordingly, rapamycin was later approved to prevent allograft rejection in transplanted patients and for use in coronary-artery stents to prevent stenosis [38,39,40]. Since rapamycin is poorly water-soluble, several analogs of rapamycin, known as rapalogs, were developed with improved pharmacokinetic properties and tested in cancer patients [41,42,43]. Overall, their anticancer effects are modest and rapalogs are prescribed in second- or third-line therapy [44]. Some mechanisms that limit the efficacy of rapalogs have been identified and include activation of alternate proliferative signaling pathways, treatment resistance mutations of mTOR and more problematic tumor heterogeneity [3].

The discovery of the rapalog-insensitive mTORC2 stimulated the development of inhibitors of mTOR that target the kinase domain. This led to the generation of adenosine triphosphate (ATP)-competitive agents to mTOR that block mTORC1 and mTORC2 [42,45]. As the kinase domains of mTOR and phosphoinositide 3-kinase (PI3K) are highly homologous, some of these inhibitors exert dual activity against mTOR and PI3K [46]. In contrast to rapalogs, mTOR kinase inhibitors totally block mTORC1 activity (Table 1). Although preclinical studies have demonstrated the anti-cancer efficacy of these inhibitors, none of these agents are currently approved for cancer therapy [44]. Of note, whereas most experimental studies demonstrated increased anticancer efficacy of the ATP-competitive inhibitor of mTOR compared to rapalogs, a phase 2 trial in renal cell carcinoma patients reported that AZD2014, a kinase inhibitor of mTOR, was less efficient than the rapalog everolimus [47,48,49]. Ongoing trials will help identify the future role that kinase inhibitors of mTOR will endorse in cancer therapy.

A third generation of mTOR inhibitors was designed based on the analysis of mTOR mutations that provide resistance to rapalogs or kinase inhibitors of mTOR [50]. In fact, two types of mutations were observed. Firstly, FRB domain mutations that disrupt the interaction of mTOR with FKBP12-rapamycin complex. Secondly, mutations of the kinase domain that generate a hyperactive kinase state without blocking the binding of kinase inhibitors to the domain. To overcome these mutations, a molecule named RapaLink-1 was synthesized that consists of rapamycin linked to a kinase inhibitor of mTOR and that allows inhibition of the mutants [50]. Compared to rapamycin or to the kinase inhibitor of mTOR sapanisertib, RapaLink-1 provides a stronger anti-cancer effect in different glioblastoma models [51]. In particular, whereas orthotopic glioblastoma xenografts grow steadily under rapamycin or sapanisertib treatments, RapaLink-1 produces an initial tumor regression followed by tumor stabilization. In addition, RapaLink-1 targets glioblastoma stem cells and potentiates the anti-cancer efficacy of temozolomide, further supporting a therapeutic potential of RapaLink-1 in glioblastoma patients [52]. The anticancer benefit of RapaLink-1 is, however, not restricted to glioblastoma, as it also reduces the growth of sunitinib-resistant renal cell carcinoma tumor xenograft and prostate cancer patient-derived tumor xenograft [53,54].

Finally, since mTORC2 participates in cancer progression, specific mTORC2 inhibitors could be of therapeutic benefit for cancer patients [55]. Compared to kinase inhibitors of mTOR, specific mTORC2 would not induce compensatory cell survival and proliferation induced by mTORC1 inhibition. Recently, a small molecule that blocks mTOR–RICTOR interaction was identified using a high-throughput yeast two-hybrid screen [56]. Besides blocking mTORC2 activity, it reduced glioblastoma xenograft growth, supporting the use of such compound to fully identify the role played by mTORC2 in cancer and to characterize the therapeutic potential of its inhibition.

## 3. Immunomodulatory Effects of mTOR Inhibitors on T Cells

Rapamycin was initially evaluated as an anti-fungal drug [57]. However, this was quickly abandoned following the discovery of the immunosuppressive effects of rapamycin (Figure 2) [36]. These effects were then further demonstrated in models of autoimmunity, graft versus host disease and organ transplantation, that ultimately led to the food and drug administration (FDA) approval in 1999 of rapamycin as an immunosuppressant to prevent allograft rejection [4,58,59,60,61]. Interestingly, the immunosuppressive activity of rapamycin was later challenged by the observation that transplanted patients treated with rapalogs had a reduced risk of cytomegalovirus infection [62,63,64]. Since then, several studies have demonstrated that rapalogs act also as immuno-stimulators, and for instance, improve immune functions in cancer patients and in the elderly [65,66,67]. In addition, much has been learned about the complex roles played by mTORC1 and mTORC2 in immune cells and it is now clear that targeting mTOR provides both immunosuppressive and immunostimulatory effects [5,68,69]. In fact, mTOR regulates immune cell differentiation and function by acting as a major metabolic sensor [68,70,71,72].

Initially, the immunosuppressive effects of rapamycin were attributed to the inhibition of interleukin-2 (IL-2)-mediated T cell proliferation [74,75]. Subsequent investigations suggested that this was not due to T cell proliferation inhibition but rather to biochemical events that occur during progression from the G1 into the S phase of the cell cycle [76]. It was later observed that rats treated with therapeutic doses of rapamycin had an increased proportion of regulatory CD4^+^ T cells (Tregs) in spleen, lymph nodes and bone marrow [77]. In addition, activation of CD4^+^ T cells in the presence of rapamycin resulted in the selective expansion of Tregs that were able to suppress allograft rejection [78]. The relevance of these findings was further confirmed in humans, where rapamycin promoted Tregs development from naïve T cells [79]. Likewise, a dual PI3K/mTOR exerted similar effects on Tregs differentiation and function as rapamycin [80]. Additional genetic studies revealed the complex role played by mTOR signaling in Tregs. Indeed, whereas deletion of mTOR in CD4^+^ T cells promoted CD4^+^ T cell differentiation to Tregs, loss of mTORC1 activity in Tregs impaired Tregs suppressive function and led to autoimmunity [81,82]. Furthermore, overactivation of mTORC1 or mTORC2 negatively impacts Tregs. Deletion of TSC1 in Tregs, which results in mTORC1 hyperactivation, reduces Foxp3 expression and converts Tregs in effector T cells [83]. Similarly, overactivation of mTORC2 following phosphatase and tensin homolog (PTEN) ablation in Tregs results in loss of foxp3 expression and development of autoimmune disease [84]. Therefore, from a therapeutic perspective, Tregs can be inhibited by either too low activity of mTORC1 or too high activities of mTORC1 or mTORC2 [70,82,85].

Immunostimulatory activity of rapamycin was first demonstrated in infection and later in cancer models, where rapamycin stimulated the generation of memory CD8^+^ T cells [86,87]. Genetic ablation, specifically in CD8^+^ T cells, of components of the mTORC1 signaling pathway confirmed, in part, these observations. For instance, CD8^+^ T cells deficient in mTORC1 activity following Rheb deletion failed to differentiate into effector cells but generated more memory T cells [88]. However, these memory T cells were not able to respond to secondary immunization, highlighting the importance of mTORC1 in memory T cell recall activity. In contrast, constitutive activation of mTORC1 via tuberous sclerosis complex subunit 2 (TSC2) deletion in CD8^+^ T cells promoted the development of effector CD8^+^ T cells and abrogated the generation of memory T cells [88]. Similar observations were reported by selectively overexpressing RHEB or PRAS40 in CD8^+^ T cells to increase or reduce mTORC1 activity, respectively [89]. Hence, while promoting memory CD8^+^ T cell differentiation, blocking mTORC1 also impairs memory CD8^+^ T cell function and effector CD8^+^ T cell expansion. Therefore, from the perspective of cancer therapy, it will be crucial to identify the dosage and administration schedule of mTOR inhibitors necessary to favor memory CD8^+^ T cell generation without affecting effector expansion [70]. Of note, the development of memory CD8^+^ T cells is also increased in mTORC2-deficient CD8^+^ T cells, suggesting that mTORC2 inhibitors represent a therapeutic option that needs further investigations [88]. Finally, the immunostimulatory effects of mTOR inhibitors are not restricted to CD8^+^ T cells since rapamycin also enhances the cytotoxic effects of γδ T cells, enhancing their antitumor efficacy [90,91,92].

Although this review focuses on chemical inhibition of mTOR, it is worth mentioning that therapeutic approaches using genetic ablation of components of mTOR complexes have been developed, allowing mTOR inhibition in specific T cell subsets, therefore avoiding undesirable immunosuppressive effects of pharmacological agents. For example, conjugation of a raptor targeting siRNA to an aptamer that binds 4-1BB (CD137) specifically inhibits mTORC1 activity in activated CD8^+^ T cells. Consequently, the 4-1BB aptamer-raptor siRNA conjugate was superior to systemic rapamycin in terms of protective antitumor immunity [93]. In particular, rapamycin, but not the aptamer-targeted siRNA, reduced the cytotoxic effector functions of reactivated memory CD8^+^ T cells and diminished alloreactivity of dendritic cells. Also, as mentioned above, whereas mTORC1 activity is required for the initial effector response, mTORC1 inhibition favors memory differentiation. Hence, a therapeutic approach using 4-1BB aptamer-raptor siRNA could be particularly beneficial, where cells with greater inhibition levels would differentiate in memory precursor and cells with low inhibition would participate in the initial effector response [89].

## 4. Combining mTOR Inhibitors and Immunotherapies in Cancer Therapy

Several preclinical studies have highlighted the anticancer benefits of combining mTOR inhibitors with immunotherapies. In this section, we provide an overview of these studies and put a particular emphasis on experimental protocols, as finding the appropriate dosage and administration schedule of mTOR inhibitors will be critical to shift mTOR inhibitors-mediated immune response toward antitumor immunity.

### 4.1. mTOR Inhibitors and Antitumor Vaccines

Based on the observations that rapamycin enhances the efficacy of vaccines targeting bacteria or virus, the effect of mTOR inhibitors on vaccines targeting cancer was investigated (Table 2) [86,94]. The rapalog temsirolimus increased the anticancer efficacy of heat-shock protein-based vaccines in established renal cell carcinoma and melanoma in mice [95]. In these experiments, vaccines were administered 10 days and 17 days post-tumor cell inoculation and 15 μg of temsirolimus was given daily from day 11 to the end of the experiment. In addition, in a tumor prevention protocol, a single dose of tumor vaccine was injected on day 0 followed or not by temsirolimus treatment from day 8 to day 32, and melanoma tumor cell inoculation on day 150. Whereas administration of vaccine alone significantly decreased tumor growth, administration of vaccine and temsirolimus prevented the growth of tumors in all mice over 60 days [95]. These latter results suggest that temsirolimus given during the contraction phase favors transition from effector CD8^+^ T cells to memory cells consistent with other observations that demonstrated that rapamycin treatment during CD8^+^ T cell contraction increases CD8^+^ T cell memory differentiation [86].

Other models were used to further demonstrate that rapalogs enhance the efficiency of antitumor vaccines. For instance, mice bearing B16 tumors expressing ovalbumin (OVA) were vaccinated 14 days post-tumor inoculation with the shigella dysenteriae B-subunit toxin based (STxB-OVA) vaccine, followed by temsirolimus treatment (2 mg/kg) three days later and on day 20 and 26. Tumor growth was significantly decreased by vaccination combined with temsirolimus compared to vaccination or temsirolimus alone [96]. This effect was associated with a higher percentage of OVA-specific CD8^+^ T cells in the tumor microenvironment. CD8^+^ T cell depletion markedly reduced the anticancer efficacy of the combined treatment, thus confirming the contribution of CD8^+^ T cells. Finally, OVA-specific CD8^+^ T cells’ phenotype analysis revealed that vaccination combined with temsirolimus induced high expression of CD127 and CD67L and low expression of KLRG1 hallmarks of precursor and central memory T cells. Similar results were obtained with an E7-peptide-based vaccine in a TC-1 tumor model [96]. Interestingly, it was previously shown that mTOR inhibitors also increase Tregs, which dampens their anti-tumor response [97]. This was further highlighted in both B16-OVA and TC-1 models as depletion of Tregs or Tregs blockade using a CCR4 antagonist further delayed tumor growth in mice treated with vaccine and temsirolimus [96].

A positive impact of mTOR inhibition was also noted following vaccination with antigen-encoding RNA [98]. In this case, rapamycin (600 μg/kg) administered during the contraction phase had no effect on the absolute number of circulating, antigen-specific CD8^+^ T cells with decreased effector CD8^+^ T cells but increased the number of antigen-specific memory CD8^+^ T cells. In contrast, when rapamycin was given during the priming and expansion phase, the number of antigen-specific CD8^+^ T cells was markedly decreased [98]. This translated into improved anti-tumor efficacy of the RNA vaccine when combined with rapamycin administration during the contraction phase compared to RNA vaccine or rapamycin alone. Combining vaccine and rapamycin did not significantly augment the total number of CD8^+^ T cells and natural killer (NK)cells in tumors compared with RNA vaccination alone. In contrast, the frequency of antigen-specific CD8^+^ T cells was increased in tumors of mice treated with rapamycin and RNA vaccine.

**Table 2 cancers-13-01359-t002:** mTOR inhibitors and immunotherapies in preclinical cancer models.

Tumor Model	Approaches	Results	Ref.
Vaccines			
RCC RENCAMelanoma B16	- Tumor cell inoculation (day (d) 0) - HSP-based vaccine (d10 and d17) - Temsirolimus (d11–d24)	Temsirolimus increases the anticancer effect of HSP vaccine.	[95]
Melanoma B16	- HSP-based vaccine (d0) - Temsirolimus (d8–d32) - Tumor challenge (d150)	Combining HSP vaccine and temsirolimus prevents tumor growth.	[95]
Melanoma B16 OVA	- Tumor cell inoculation (d0) - STxB-OVA vaccine (d14) - Temsirolimus (d17–d20–d26)	- Combining vaccine and temsirolimus reduces tumor growth. - Combined therapy increases OVA-specific CD8^+^ T cells in tumors. - Combined therapy increases memory T cell generation. - Tregs blockade increases efficacy of combined therapy.	[96]
TC-1 tumor	- Tumor cell inoculation (d0) - E7 peptide vaccine (d10 and d20) - Temsirolimus (d14–d16–d18–d22–d26)	- Combining vaccine and temsirolimus reduces tumor growth. - Combined therapy increases E7-specific CD8^+^ T cells in tumors. - Tregs blockade increases efficacy of combined therapy.	[96]
Melanoma B16 OVA	- Tumor cell inoculation (d0) - RNA based vaccine (d3–d6–d9) - Rapamycin (d11–d33)	- Combining vaccine and rapamycin decreases tumor growth. - Combined therapy increases OVA-specific CD8^+^ T cells in tumors.	[98]
TC-1 tumor	- Tumor cell inoculation (d0) - CTGF/E7 DNA vaccine (d11–d13–d18–d20) - Rapamycin or everolimus or temsrolimus (d10–d12–d17–d19)	- Combining vaccine and rapalogs decreases tumor growth. - Everolimus and rapamycin increases E7-specific CD8^+^ T cells in draining lymph nodes.	[99]
Melanoma B16 OVA	- Tumor cell inoculation (d0) - Dendritic cell vaccine pulsed with or without rapamycin (d3)	Vaccination with rapamycin treated dendritic cells reduces tumor growth and increases OVA-specific CD8^+^ T cells in tumors.	[100]
EG.7-OVA thymoma cells	- Vaccination with LM-OVA (d1) - Anti-CTLA-4 (d1) - Rapamycin (d1–d10) - EG.7 tumor cell challenge (d50)	Combining vaccine with rapamycin and anti-CTLA-4 decreases tumor growth and increases OVA-specific CD8^+^ T cells in the blood.	[101]
EG.7-OVA thymoma cells	- Tumor cell inoculation (d0) - Tricom vaccination (d10) - Rapamycin high or low dose (d0–d7 or d0–d39)	Combining vaccine with rapamycin high dose (d0–d7) decreases tumor growth.	[102]
TC-1 tumor cells	- Tumor cell inoculation (d0) - CyaA-E7 vaccination (d14) - Rapamycin high or low dose (d0–d22 or d14–d22)	Rapamycin abrogates antitumor effect of vaccination.Rapamycin decreases CD8^+^ T cell recruitment in tumors.Rapamycin increases Tregs suppressive function in tumors.	[103]
Melanoma B16	- GVAX vaccination (d0) - Rapamycin (d1) - Tumor challenge (d50)	No effect of rapamycin	[93]
Immune checkpoint modulators			
MOC1 tumor cells	- Anti-PD-L1 (d1–d2–d3 once tumor reaches 0.1 cm^3^) - Rapamycin (d1–21)	Rapamycin increases anticancer effect of anti-PD-L1 Ab	[104]
MC38 tumor cells	- -Tumor cell inoculation (d0) - Anti-PD-1 or anti-PD-L1 or anti-CTLA-4 Abs (d1 followed by twice a week) - Vistusertib (d1–end of experiment)	- Vistusertib potentiates the antitumor effect of checkpoint inhibitors - Vistusertib decreases exhaustion phenotype of TILs	[105]
CT-26 tumor cells	- -Tumor cell inoculation (d0) - Anti-PD-1 or anti-PD-L1 or anti-CTLA-4 Abs (d1 followed by twice a week) - Vistusertib (d1–end of experiment)	- Vistusertib potentiates the antitumor effect of checkpoint inhibitors - Vistusertib decreases exhaustion phenotype of TILs	[105]
RCC RENCA	- Tumor cell inoculation (d0) - Anti-CD40 Ab (d3–d7 and d10–d14 and d16) - AZD8055 (d7–d10 and d13–d16)	- Antitumor effect of anti-CD40 Ab increases when administered prior to AZD8055 - Combined therapy increases activated CD8^+^ T cells, dendritic cells, macrophages and Tregs in tumors	[106]
Hepatocellular carcinoma tumor xenografts	- Tumor cell inoculation (d0) - Anti-PD-1 Ab and INK-128 administered simultaneously (d0–end of experiment)	INK-128 potentiates the anticancer effects of anti-PD1 Ab via a non-immune-related mechanism	[107]
Adoptive cell transfer			
EG.7 OVA thymoma cells	- Injection of OT-I cells preconditioned or not with rapamycin (d0) - Tumor cell challenge (d1)	Anticancer efficacy of rapamycin-treated OT-I cells is increased compared to untreated cells.	[87]
EG.7 OVA thymoma cells	- Tumor cell inoculation (d0) - Injection of naïve OT-I cells (d7) - Rapamycin (d7–d14)	Rapamycin increased antitumor effects of OT-I cells	[108]
Melanoma B16F10	- Tumor cell inoculation (d0) - Injection of Vγ4γδ T cells treated ex-vivo with rapamycin (d1–d3–d5–d7–d9–d11)	Antitumor effect of rapamycin-treated Vγ4γδ T cells is increased compared to untreated cells	[90]
Carcinogen-driven skin cancer	- Tumor induction (d0) - Injection of γδ T cells isolated from mice treated or not with rapamycin (d20) - Rapamycin (d20–d34)	Antitumor effect of γδ T cells is increased by pre-exposition to rapamycin	[92]

Besides antigen-encoding RNA, mTOR inhibitors also enhance the anti-tumor effects of DNA vaccines [99]. TC-1 tumor cells were injected into C57BL/6 mice on day 0. Ten days later, mice were treated or not with different rapalogs, including rapamycin (1 mg/kg), everolimus (1 mg/kg) or temsirolimus (5 mg/kg) twice a week for two weeks. On day 11, connective tissue growth factor (CTGF/E7) was administered or not also twice a week for two weeks. Mice survival was significantly increased in the groups that received the vaccine and rapalog compared to either treatment alone or no treatment. Of note, treatment with vaccine and everolimus or rapamycin significantly increased E7-specific interferon-γ (IFN-γ)-secreting CD8^+^ T cells in tumor-draining lymph nodes.

Targeting mTOR also increases the antitumor effects of dendritic cell vaccines [100]. Mice were inoculated with OVA-expressing B16 melanoma cells and immunized three days later with dendritic cells that were previously pulsed for 6 h with lipopolysaccharides (LPS), with LPS and OVA or with LPS and OVA and rapamycin. Tumor growth was significantly reduced in mice vaccinated with rapamycin-treated dendritic cells compared with dendritic cells treated with LPS and OVA or LPS alone. Analysis of harvested tumors revealed that OVA-specific CD8^+^ tumor-infiltrating lymphocytes were significantly increased following immunization with rapamycin-treated dendritic cells [100]. In vitro, rapamycin prolonged dendritic cell lifespan and enhanced expression of CD40 and CD86 costimulatory molecules. Rapamycin inhibited AKT phosphorylation in dendritic cells, a downstream effector of mTORC2, suggesting that inhibition of mTORC2 might be involved in the effects of rapamycin on dendritic cells. Consistent with this hypothesis, intra-tumoral injection of RICTOR-deficient dendritic cells slowed B16 melanoma tumor growth [109]. Again, this effect was associated with increased frequencies of cytotoxic CD8^+^ tumor-infiltrating lymphocytes.

The effect of rapamycin on the efficacy of anti-cancer vaccination was also tested in combination with cytotoxic T-lymphocytes-associated protein 4 CTLA-4 blockade [101]. Mice were immunized with recombinant OVA expressing Listeria monocytogenes in the presence of a single dose of anti-CTLA-4 and 10 days of low-dose rapamycin (75 μg/kg/day) injections. Fifty days later, immunized animals were challenged with EG.7 cells (EL4 tumor cells expressing OVA Ag) to test memory-mediated anti-tumor response. A significant decrease in tumor growth and significant increase in tumor-free survival were observed in mice that received both anti-CTLA-4 and rapamycin during T cell priming compared to either treatment alone. This effect was associated with a significant increase in the frequency of antigen-specific CD8^+^ T cells in peripheral blood leucocytes.

The effects of rapamycin dose and treatment duration on the efficacy of antitumor vaccine have also been partially addressed in preclinical studies [102]. Mice bearing EG.7-OVA thymoma tumors were vaccinated on day 0 with the canary poxvirus expressing chicken OVAmLFA-3/mICAM/mB7.1 (Tricom) 10 days after tumor cell inoculation. Rapamycin (low dose: 0.075 mg/kg/day, or high dose: 0.75 mg/kg/day) was given by intraperitoneally (i.p.) injection either from day 0 to 7 or day 0 to 39. Of the different treatment regimen, only rapamycin high dose from day 0 to 7 increased the anti-cancer efficacy of Tricom vaccination.

It is important to mention that negative effects of mTOR inhibitors on the anti-tumor response induced by vaccination were reported [103]. Here, C57BL/6 mice were grafted with TC-1 tumor cells on day 0. CyaA-E7 vaccination with CpG-B adjuvant was administered on day 14 and mice were treated with rapamycin (750 μg/kg/day) either from day 0–22 or from day 14–22. Vaccination induced full tumor regression in 42% of mice and significantly enhanced mice survival. This therapeutic effect was, however, totally abrogated by either regimen of rapamycin. Similar suppressive effects were obtained when a low dose of rapamycin (75 μg/kg/day) was used. Additional analysis revealed that high doses of rapamycin inhibited tumor- and vaccine-induced CD8 T cell recruitment to the tumor site. In contrast, rapamycin enhanced the intra-tumoral recruitment of myeloid-derived suppressor cells and increased suppressive function of Tregs in tumors. In addition, rapamycin did not increase the memory phenotype of CD8 T cells in lymphoid tissues.

Absence of effects of rapamycin on anti-tumor vaccination was also observed. In a prophylactic experimental design, mice were vaccinated with granulocyte-macrophage colony-stimulating factor (GM-CSF) expressing irradiated B16 melanoma cells (GVAX) on day 0 and treated with rapamycin 24 h later (15 μg, three times daily, i.p.). On day 50, mice were inoculated with B16 melanoma cells and tumor growth was monitored [93]. Under these experimental settings, rapamycin did not increase mice survival compared to untreated mice.

Dose and schedule of administration of mTOR inhibitors are important to tip the balance toward immunostimulatory effects. All studies showing benefits of combining anti-tumor vaccines with mTOR inhibitors followed a protocol where mTOR inhibitors were given after or on the same day as vaccination. High or low doses were used with no clear benefit of one regimen over the other. Nevertheless, one study reported detrimental consequences of adding rapamycin, either high or low doses, to antitumor vaccine, highlighting the need to further clarify and identify appropriate therapeutic protocols that are probably not limited to dose and schedule [103,110]. For instance, the model used might also influence the effect generated by mTOR inhibitors. It was indeed reported that rapamycin increases OVA-specific CD8^+^ T cell response in mice induced by infection of OVA-expressing bacteria but not by OVA-expressing skin grafts [111]. One important point to address related to rapalogs dosage is whether they fully or partially inhibit mTORC1 and whether they inhibit only mTORC1 or both mTORC1 and mTORC2. Given that mTORC2 also participates in generation of memory T cells, the differential inhibitory effects of rapalogs on mTOR complexes also affect the immune response and might further explain the discrepancies observed between the different experimental models [88].

### 4.2. mTOR Inhibitors and Modulation of Immune Checkpoints

Potentiating benefits of mTOR inhibitors in combination with modulators of immune checkpoints have been reported in experimental tumor models (Table 2). For instance, mice bearing MOC1 tumors had prolonged survival when treated with rapamycin in combination with anti-PD-L1 mAb compared to either treatment alone [104]. Treatments were started once tumors had reached 0.1 cm^3^ with three doses of anti-PD-L1 antibody and a loading dose of rapamycin (4.5 mg/kg), followed by injections every other day of 1.5 mg/kg for 21 days. Flow cytometry analysis of isolated tumors revealed that anti PD-L1 treatment increased infiltration of antigen-specific CD8^+^ T cells as well as activated antigen-specific CD8^+^ T cells, which was preserved with the addition of rapamycin. Ex vivo analysis of CD8^+^ tumor-infiltrating lymphocytes (TILs) showed that the ability of CD8^+^ TILs to produce IFN-γ following stimulation with PMA/ionomycin was significantly enhanced by rapamycin in combination with anti-PD-L1 mAb compared to monotherapies. The number of infiltrating Tregs or myeloid-derived suppressor cells (MDSC) was not significantly modified by the different treatments. Similar to CD8^+^ TILs, combination of anti-PD-L1 mAb with rapamycin increased tumor-infiltrating NK cells. However, depletion of CD8 and not NK abolished the anti-cancer effects generated by rapamycin and anti-PDL1 mAb, highlighting the role of CD8^+^ T cells in this process [104].

Similar to rapamycin, targeting mTOR with kinase inhibitors potentiates the anti-cancer effects of checkpoint inhibitors. Indeed, combining mTORC1/mTORC2 inhibitor vistusertib with different immune-checkpoint blockades, including anti-PD-1, anti-PD-L1 or anti-CTLA-4, significantly reduced MC38 or CT-26 tumor growth compared to monotherapies [105]. Treatments were started 4 days (CT26) or 1 day (MC38) after tumor cell implantation and consisted of daily administrations of vistusertib (15 mg/kg) and twice a week administrations of checkpoint inhibitors. Overall, compared to single immune checkpoint blocking antibodies, addition of vistusertib decreased the exhaustion phenotype of TILs. In addition, the vistusertib/anti-CTLA-4 combination increased IFN-γ−expressing CD8^+^ T cells in CT26 tumors [105].

Anti-cancer benefits were further generated when the mTOR kinase inhibitor AZD8055 was combined with a CD40 agonistic antibody in a model of metastatic renal cell carcinoma [106]. In this model, RENCA renal cancer cells are injected intra-splenically and the resulting liver tumor nodules are counted 17 days later. Both CD40 agonistic antibody and AZD8055 reduced liver nodules, and this effect was, however, significantly increased when both treatments were used in combination. Interestingly, the effect was maximum when CD40 agonistic antibody was given prior to AZD8055. In contrast, no enhancing effect was observed when CD40 agonistic antibody was administered with rapamycin. AZD8055/CD40 agonistic antibody significantly increased tumor infiltration by activated CD8^+^ cells, dendritic cells and macrophages compared to either treatment alone. Of note, the combination treatment also increased the number of Tregs in the liver.

Finally, potentiating benefits of combining mTOR inhibitors with anti-PD-1 antibody have been reported in a model of hepatocellular carcinoma, however independently of adaptive immunity [107]. Indeed, combining the mTOR kinase inhibitor INK128 with PD-1 blockade significantly decreased hepatocellular carcinoma growth generated in immunosuppressed NOD/SCID mice compared to single treatments. In fact, certain cancer cells express PD-1 which induces proliferative signaling following activation via a non-immune-related process [107,112].

Taken together, these studies demonstrate the anticancer benefits of combining mTOR inhibitors with immune checkpoint modulators. Complete mTORC1 and mTORC2 inhibition seems necessary as high doses of rapamycin or kinase inhibitors of mTOR were used. Interestingly, clinical trials are already exploring the anticancer efficacy of such an approach in cancer patients (NCT02423954, NCT02890069, NCT04348292).

### 4.3. mTOR Inhibitors and Adoptive T Cell Transfer

The ability of mTOR inhibitors to favor differentiation of specific T cell fates provides potential benefit to adoptive T cell transfer (Table 2). For instance, adoptive transfer of IL-12-conditioned OT-I cells that were pretreated with rapamycin showed increased anti-cancer effects compared to untreated ones [87]. Similarly, rapamycin increases homeostatic proliferation-mediated CD8^+^ T cell tumor immunity [108]. In this case, radiated mice bearing EG.7 thymoma tumor cells received naïve OT-I cells seven days post-tumor cell inoculation. Rapamycin (0.75 mg/kg/day) was injected or not for 7 days starting from the day of OT-I cell transfer. The benefits of rapamycin in the context of adoptive T cell transfer are not limited to CD8^+^ T cells and have also been documented for γδ T cells. B16-F10 tumor growth was significantly more decreased following injection of Vγ4γδ T cells that were treated ex vivo with rapamycin compared to untreated Vγ4γδ T cells [90]. In vitro analysis revealed that rapamycin increased the cytotoxic activity of Vγ4γδ T cells. Rapamycin also increased the anticancer efficacy of γδ T cells in a carcinogen-driven skin cancer model [92]. Here, γδ T cells were isolated from skin-draining lymph nodes of mice treated or not with rapamycin and injected in skin tumors. Tumor growth was significantly reduced by γδ T cells that were isolated from rapamycin-treated mice compared to untreated mice.

### 4.4. mTOR Inhibitors and Chimeric Antigen Receptor (CAR) T Cells

CAR T cells have demonstrated antitumor activities, in particular in hematological malignancies [113,114]. The success of CAR T cells relies in particular on the ex vivo culture conditions, as injection of less differentiated CAR T cells is associated with prolonged anti-tumor response [115]. In this context, expansion of CAR T cells in the presence of IL-15 results in the generation of less differentiated CAR T cells with increased anti-tumor activity, presumably due to reduced mTORC1 activity [116]. To support this hypothesis, CAR T cells expanded in the presence of IL2 and rapamycin display a similar phenotype [116]. Hence, mTOR inhibition might provide benefits during ex-vivo manufacturing of CAR T cells. Additional experiments are also necessary to address the therapeutic potential of treating patients with mTOR inhibitors during the infusion of CAR T cells.

## 5. Immune Effects of mTOR Inhibitors in Cancer Patients

A few studies have investigated the immune effects of mTOR inhibitors in cancer patients and have confirmed the bimodal activity observed in murine models. For instance, an immuno-monitoring study performed in 23 metastatic renal cell carcinoma patients revealed that rapaologs increased the percentage and absolute number of T regs in 21 of 23 patients [8]. Rapalogs concomitantly induced a Th1 tumor-specific response in 6 patients and disease progression was associated with reduced anti-tumor Th1/Tregs ratio. A shift of the balance toward an immunosuppressive state was reported in 5 renal cell carcinoma patients, where everolimus increased the percentage of Tregs, increased myeloid-derived suppressor cells and decreased the activation of several dendritic cell subtypes [117]. Of note, these studies monitored circulating immune cells which might therefore not fully reflect the immune status of the tumor microenvironment. However, acquisition of tumor samples following treatment is difficult to justify in these patients. Nevertheless, these studies suggest that combining mTOR inhibitors with therapies that target Tregs might provide therapeutic benefits. In this context, a clinical trial is evaluating the anti-cancer effect of everolimus combined to metronomic low doses of cyclophosphamide known to selectively deplete Tregs [118]. Finally, immuno-protective effects of rapamycin were observed in bladder cancer patients undergoing cystectomy. Surgery induced T cell exhaustion, as evidenced by an increased proportion of circulating T cells expressing markers of exhaustion including PD-1, Tim-3 and Lag-3, which was prevented by rapamycin treatment prior to surgery [66].

## 6. Conclusions

The use of mTOR inhibitors in cancer therapy did not provide the expected antitumor benefits. Several resistance mechanisms that counteract their efficacy have been identified and have highlighted the need to use mTOR inhibitors in combination therapies. In this context, different studies have demonstrated that mTOR inhibitors provide anticancer benefits to immunotherapy. In particular, the ability of mTOR inhibitors to promote memory CD8^+^ T cells represents a major advantage that needs, however, to be carefully weighed against their immunosuppressive effects [119].

Despite the proven anticancer benefits of combined mTOR inhibitors and immunotherapy in preclinical models, several challenges exist when applying such treatment protocols in cancer patients. Firstly, precise conditions related to dosage and administration schedule need to be characterized as little changes in mTORC1 or mTORC2 activities can have profound impact on the immune response and promote anergic/tolerogenic effects instead of memory ones [119]. In fact, long-term administration of mTOR inhibitors exerts a dual effect on patient antitumor immune responses and designing protocols that tip the balance towards immuno-stimulation will be critical [8]. Alternatively, inhibition of the immunosuppressive effect of mTOR inhibitors represents another treatment option that has already been validated in tumor vaccine experimental settings [96]. Secondly, whereas dose- and time-dependent experiments are easily performed in mice models, such experiments require major resources in patients. In addition, since mTOR inhibitors exert different immunomodulatory effects between patients, personalized treatment regimens will be needed. Hence, development of such protocol might fail due to economic issues. In this context, the use of mTOR inhibitors during the de novo expansion of immune cells for adoptive cell therapy would be easier to set in place. Thirdly, the effects of mTOR inhibitors on patient immunity are complex and involve almost every immune cell [8,117]. Consequently, a complete monitoring of the immunostimulatory and immunosuppressive effects of mTOR inhibitors will be difficult to achieve in practice. Fourthly, if a precise regimen of mTOR inhibitors’ administration is required, patient adherence to medication will be essential. This is particularly problematic with oral anticancer medication such as everolimus, with reported adherence rates ranging from less than 20% to 100% [120]. Hence, interventions to ensure patient adherence will be required. If not possible, intravenous drug administration could also be preferred.

Nearly forty years after the discovery of the immunosuppressive effects of rapamycin, clinical studies have been launched to explore the immunostimulatory benefits of mTOR inhibitors. For instance, a phase Ib clinical trial is investigating the effect of rapamycin and the anti-PD-L1 antibody durvalumab in patients with stage I–IIIA non-small lung cell cancer (Clinicaltrials.gov: NCT04348292). There is no doubt that such trial will shed light on the immunostimulatory benefits of mTOR inhibitors in cancer patients and if successful, will set the stage for a new beginning for mTOR inhibitors in cancer therapy.

## Figures and Tables

**Figure 1 cancers-13-01359-f001:**
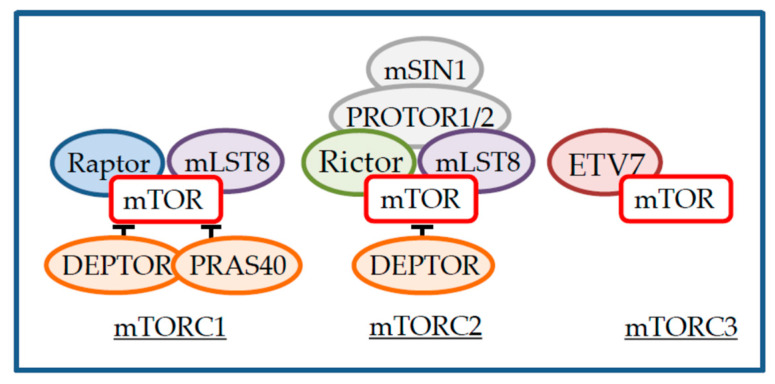
Components of mechanistic target of rapamycin (mTOR) complexes.

**Figure 2 cancers-13-01359-f002:**
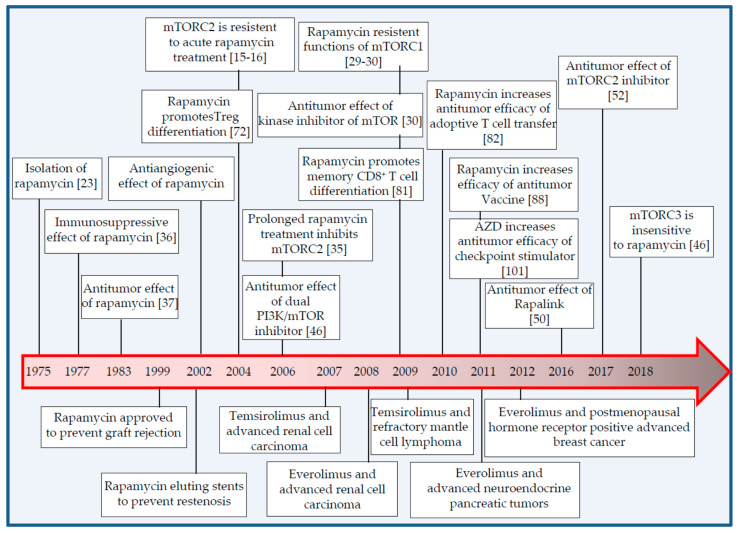
Milestones in development and therapeutic applications of mTOR inhibitors in the field of malignant cancers and immunity. Major therapeutic applications are described below the timeline. Corresponding studies are provided in square parentheses. Adapted from References [42,73].

**Table 1 cancers-13-01359-t001:** mTOR inhibitors and their effects on mTOR complexes.

mTOR Inhibitors		mTORC1	mTORC2	mTORC3	PI3K
**Approved**					
Rapalogs Rapamycin (sirolimus), RAD001 (everolimus), CCI-779 (temsirolimus)					
	Short-term	+ partial	−	−	−
	Long-term	+ partial	+ cell-dependent	−	−
	High concentrations (μM)	+	+	n/a	−
**In clinical trials**					
Kinase inhibitors of mTOR AZD2014 (vistusertib), INK-128 (sapanisertib), AZD8055					
		+	+	+	−
Dual PI3K/mTOR inhibitors NVP-BEZ235 (dactolisib), PQR309 (bimiralisib), PKI587 (Gedatolisib)					
		+	+	n/a	+
**In preclinical development**					
RapaLink-1		+	+	n/a	−
JR-AB2-011		−	+	n/a	−

(+): Inhibition; (−): no inhibition; n/a: data not available.

## Data Availability

Not applicable.

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
