# Peer review of "Combining mTOR Inhibitors and T Cell-Based Immunotherapies in Cancer Treatment"

_cancers, 2021, doi:10.3390/cancers13061359_

Round 1

Reviewer 1 Report

The review is very well organied and present a good reading for researchers interested in the comnination of mTOR inhibitors and immuno-target strategical therapies for cancer.  The detailed comments are in the attachement.

Author Response

We are grateful to you for your nice comments and for the time spent to review our manuscript

Reviewer 2 Report

In the manuscript: “Combining the mTOR inhibitors and immunotherapies in cancer treatment” the authors describe mTOR inhibitors and their effects on the preclinical efficacy of immunotherapies, mainly those dependent on T cells. This is a very important, timely but also challenging topic. The mTOR inhibitors exert very divergent effects on immune cells and immune response, from immunosuppressive to immunostimulatory, which makes the topic very complex.

In my opinion, the review would be more impactful and useful for the readers if the authors attempted to better order the existing research data, provide more illustrative figures/tables, and interpret/discuss the results of the preclinical data more accurately.

Specific comments:

  1. As the review focuses almost exclusively on T cell-based immunotherapies, it should be highlighted in the title. For example, the role of mTOR in NK cells and hence NK cell-dependent immunotherapies are not discussed.
  2. Section 2:
  • a figure illustrating the mTOR complexes, their activators and the key cellular processes regulated by the complexes would be useful; alternatively, please provide a reference to a corresponding figure in the literature;
  • Table 1 should include specific inhibitor names, the most important once (all clinically approved and tested in clinics) as well as all these that are mentioned in the review; maybe also the reorganization of the table could be considered, to include the information of the stage of development for each individual inhibitor (preclinical, clinical trials or approvals (in which indications);
  • more information on the anticancer efficacy of rapalink should be provided.
  1. Section 3:
  • Figure 1 was presumably inspired by Fig 2 published in Ni Bhaoighill, et al 2019. Mechanistic target of rapamycin inhibitors: successes and challenges as cancer therapeutics. Cancer Drug Resistance2019 (2) , pp. 1069-1085. 20517/cdr.2019.87. This should be clearly stated. Moreover, the figure is unclear, it is not highlighted that the lower panel includes FDA approvals; more recent approvals, after 2012 should also be included;
  • the paragraph describes the effects of mTOR inhibitors on T cells only; it should be stated in the paragraph subtitle;
  • it would be useful to include the figure presenting immunosuppressive and immunostimulating effects of mTOR inhibitors;
  • PMID: 24292708 –this paper very clearly shows differences between the effects of genetic mTORC1 ablation and rapamycin on CD8 T cells, should be included in this section.
  1. Section 4:
  • The first sentence, lines 215-216 seem imprecise; promotion of memory CD8 T cell differentiation by mTORi does not mean enhancement of antitumor CD8 response, in particular that these cells are not able to respond to secondary immunization (lines 199-200);
  • The paragraph 4.1. describing mTOR inhibitors and anticancer vaccines lacks discussion of possible reasons behind the conflicting results; are there any conclusions which inhibitors/ doses/ application schemes are critical for the synergistic effects of mTORi and anticancer vaccines? Also section 4.2 would profit from additional paragraph discussing the overall results.
  • Though the Table 2 could be useful for very determined readers, it could be arranged differently, to facilitate the reader to assimilate the information more easily; in the current form it entirely overlaps with the text, maybe it could be arranges around the observed effects on different types of immune cells/ treatment modalities, with only selected examples provided for each type?
  • 3. –potential use of rapamycin in the protocols for expansion of CAR T cells could also be mentioned.
  1. Most of the discussed results are obtained in murine models. Effects of mTOR inhibitors administered systemically to cancer patients on immune cells could also be potentially discussed (e.g. 30652208, 30563829, 30451741) and would be interesting, especially in the context of combination treatments including the inhibitors and immunotherapies.

Author Response

We are grateful to reviewer 2 for her/his comments. We have carefully followed his/her recommendations to improve the quality of the manuscript

  • As the review focuses almost exclusively on T cell-based immunotherapies, it should be highlighted in the title.

We have changed the title accordingly to “ combining mTOR inhibitors and T cell-based immunotherapies in cancer treatment”

  • a figure illustrating the mTOR complexes, their activators and the key cellular processes regulated by the complexes would be useful; alternatively, please provide a reference to a corresponding figure in the literature

We have added a figure representing the complexes of mTOR and their major components. Readers can refer to ref 9, Liu and Sabatini (2020) Nat Rev Mol Cell Biol for a complete review of mTOR signaling pathway.

  • Table 1 should include specific inhibitor names, the most important once (all clinically approved and tested in clinics) as well as all these that are mentioned in the review; maybe also the reorganization of the table could be considered, to include the information of the stage of development for each individual inhibitor (preclinical, clinical trials or approvals (in which indications)

We have added compound names to the figure and classified the drugs based on their development (preclinical, clinical and approved)

  • more information on the anticancer efficacy of rapalink should be provided.

We have provided more information regarding rapalink and cancer. Also as currently most of the studies addressed the effect of RapaLink-1 we have focused our attention on this particular molecule. The following text was added to the manuscript:

“Compared to rapamycin or to the kinase inhibitor of mTOR sapanisertib, RapaLink-1 provides a stronger anti-cancer effect in different glioblastoma models [51]. In particular, whereas orthotopic glioblastoma xenografts grow steadily under rapamycin or sapanisertib treatments, RapaLink-1 produces an initial tumor regression followed by tumor stabilization. In addition RapaLink-1 targets glioblastoma stem cells and potentiates the anti-cancer efficacy of temozolomide further supporting a therapeutic potential of RapaLink-1 in glioblastoma patients [52]. The anticancer benefits of RapaLink-1 is however not restricted to glioblastoma as it also reduces the growth of sunitinib resistant renal cell carcinoma tumor xenograft and prostate cancer patient-derived tumor xenograft [53,54]. “

  • Figure 1 was presumably inspired by Fig 2 published in Ni Bhaoighill, et al 2019. Mechanistic target of rapamycin inhibitors: successes and challenges as cancer therapeutics. Cancer Drug Resistance2019 (2) , pp. 1069-1085. 20517/cdr.2019.87. This should be clearly stated. Moreover, the figure is unclear, it is not highlighted that the lower panel includes FDA approvals; more recent approvals, after 2012 should also be included;

We have added the mentioned reference as well as Benjamin et al (2011) who presented a similar figure in the legend of figure 1. We also noted that below the timeline major clinical  applications are presented. Finally, to our knowledge no other major approval in the field of malignant cancers occurred after 2012.

  • the paragraph describes the effects of mTOR inhibitors on T cells only; it should be stated in the paragraph subtitle

We have changed the subtitle of this section to “ immunomodulatory effects of mTOR inhibitors on T cells”

  • PMID: 24292708 –this paper very clearly shows differences between the effects of genetic mTORC1 ablation and rapamycin on CD8 T cells, should be included in this section

The mentioned article was included as follow “Although this review focuses on chemical inhibition of mTOR, it is worth mentioning that therapeutic approaches using genetic ablation of components of mTOR complexes have been developed allowing mTOR inhibition in specific T cell subsets, therefore avoiding undesirable immunosuppressive effects of pharmacological agents. For example, conjugation of a raptor targeting siRNA to an apatamer that binds 4-1BB specifically inhibits mTORC1 activity in activated CD8+ T cells. Consequently the 4-1BB aptamer-raptor siRNA conjugate was superior to systemic rapamycin in terms of protective antitumor immunity [93]. In particular, rapamycin but not the aptamer-targeted siRNA, reduced the cytotoxic effector functions of reactivated memory CD8+ T cells and diminished alloreactivity of dendritic cells. Also as mentioned above, whereas mTORC1 activity is required for the initial effector response, mTORC1 inhibition favors memory differentiation. Hence a therapeutic approach using 4-1BB aptamer-raptor siRNA could be particularly beneficial with cells with greater inhibition levels would differentiate in memory precursor and cells with low inhibition would participate in the initial effector response [89]”  

  • The first sentence, lines 215-216 seem imprecise; promotion of memory CD8 T cell differentiation by mTORi does not mean enhancement of antitumor CD8 response, in particular that these cells are not able to respond to secondary immunization (lines 199-200)

We agree and have deleted this sentence

  • The paragraph 4.1. describing mTOR inhibitors and anticancer vaccines lacks discussion of possible reasons behind the conflicting results; are there any conclusions which inhibitors/ doses/ application schemes are critical for the synergistic effects of mTORi and anticancer vaccines? Also section 4.2 would profit from additional paragraph discussing the overall results.

We have added the following paragraphs

“Dose and schedule of administration of mTOR inhibitors are important to tip the balance toward immunostimulatory effects. All studies showing benefits of combining anti-tumor vaccines with mTOR inhibitors followed a protocol where mTOR inhibitors were given after or one the same day as vaccination. High or low doses were used with no clear benefit of one regimen over the other, Nevertheless, one study reported detrimental consequences of adding rapamycin, either high or low doses, to antitumor vaccine highlighting the need to further clarify and identify appropriate therapeutic protocols that are probably not limited to dose and schedule [103,110]. For instance, the model used might also influence the effect generated by mTOR inhibitors. It was indeed reported that rapamycin increases OVA-specific CD8+ T cell response in mice induced by infection of OVA-expressing bacteria but not by OVA expressing skin grafts [111]. One important point to address related to rapalogs dosage is whether they inhibit fully or partially mTORC1 and whether they inhibit only mTORC1 or both mTORC1 and mTORC2. Given that mTORC2 participates also in generation of memory T cells, the differential inhibitory effects of rapalogs on mTOR complexes also affect the immune response and might further explain the discrepancies observed between the different experimental models. [88]. “

“Taken together these studies demonstrate the anticancer benefits of combining mTOR inhibitiors with immune checkpoint modulators. Complete mTORC1 and mTORC2 inhibition seems necessary as high doses of rapamycin or kinase inhibitors of mTOR were used. Interestingly clinical trials are already exploring the anticancer efficacy of such approach in cancer patients ((NCT02423954, NCT02890069, NCT04348292)”

  • Though the Table 2 could be useful for very determined readers, it could be arranged differently, to facilitate the reader to assimilate the information more easily; in the current form it entirely overlaps with the text, maybe it could be arranges around the observed effects on different types of immune cells/ treatment modalities, with only selected examples provided for each type?

We fully agree that table 2 is for very determined reader only. However our aim was to provide a complete  outlook of studies that combined mTOR inhibitors and immunotherapies. We therefore would like to keep the table as initially presented.

  • potential use of rapamycin in the protocols for expansion of CAR T cells could also be mentioned

The following paragraph was added to the manuscript

4.4 mTOR inhibitors and chimeric antigen receptor (CAR) T cells

CAR T cells have demonstrated antitumor activities in particular in hematological malignancies [113,114]. The success of CAR T cells relies in particular on the ex vivo culture conditions as injection of less differentiated CAR T cells is associated with prolonged anti-tumor response [115]. In this context, expansion of CAR T cells in presence of IL-15 results in the generation of less differentiated CAR T cells with increased anti-tumor activity, presumably due to reduced mTORC1 activity [116]. To support this hypothesis CAR T cells expanded in presence of IL2 and rapamycin display similar phenotype [116]. Hence mTOR inhibition might provide benefit during ex-vivo manufacturing of CAR T cells. Additional experiments are also necessary to address the therapeutic potential of treating patients with mTOR inhibitors during CAR T cells infusion.

  • Most of the discussed results are obtained in murine models. Effects of mTOR inhibitors administered systemically to cancer patients on immune cells could also be potentially discussed (e.g. 30652208, 30563829, 30451741) and would be interesting, especially in the context of combination treatments including the inhibitors and immunotherapies.

The following paragraph was added

“5. Immune effects of mTOR inhibitors in cancer patients

A few studies have investigated the immune effects of mTOR inhibitors in cancer patients and have confirmed the bimodal activity observed in murine models. For instance an immunomonitoring study performed in 23 metastatic renal cell carcinoma patients revealed that rapaologs increased percentage and absolute number of T regs in 21 of 23 patients [8]. Rapalogs concomitantly induced a Th1 tumor specific response in 6 patients and disease progression was associated with reduced anti-tumor Th1/Tregs ratio. A shift of the balance toward an immunossupressive state was reported in 5 renal cell carcinoma patients where everolimus increased the percentage of Tregs, increased myeloid-derived suppressor cells and decreased the activation of several dendritic cell subtypes [117]. Of note, these studies monitored circulating immune cells which might therefore not fully reflect the immune status of the tumor microenvironment. However acquisition of tumor samples following treatment is difficult to justify in these patients. Nevertheless, these studies suggest that combining mTOR inhibitors with therapies that target Tregs might provide therapeutic benefits. In this context, a clinical trial is evaluating the anti-cancer effect of everolimus combined to metronomic low-dose of cyclophosphamide known to selectively deplete Tregs [118]. Finally, immunoprotective effects of rapamycin were observed in bladder cancer patients undergoing cystectomy. Surgery induced T cell exhaustion as evidenced by an increased proportion of circulating T cells expressing markers of exhaustion including PD-1, Tim-3 and Lag-3 which was prevented by rapamycin treatment prior to surgery [66]. “

Round 2

Reviewer 2 Report

The authors responded to all suggestions.